# Classical Planning with Avoid Conditions

**Marcel Steinmetz[1], Jörg Hoffmann[1], Alisa Kovtunova[2], Stefan Borgwardt[2]**

[1] Saarland University, Saarland Informatics Campus, Saarbrücken, Germany
[2] Institute of Theoretical Computer Science, Technische Universität Dresden, Germany
lastname@cs.uni-saarland.de, firstname.lastname@tu-dresden.de

## Abstract

It is often natural in planning to specify conditions that should be avoided, characterizing dangerous or highly undesirable behavior. PDDL3 supports this with temporal-logic state constraints. Here we focus on the simpler case where the constraint is a non-temporal formula $\phi$ – the avoid condition – that must be false throughout the plan. We design techniques tackling such avoid conditions effectively. We show how to learn from search experience which states necessarily lead into $\phi$, and we show how to tailor abstractions to recognize that avoiding $\phi$ will not be possible starting from a given state. We run a large-scale experiment, comparing our techniques against compilation methods and against simple state pruning using $\phi$. The results show that our techniques are often superior.

## Introduction

It is often natural in planning to specify conditions that should be avoided. Work along these lines has so far focused on temporal-logic formulas that must be true in the state sequence induced by the plan. One prominent early approach used such formulas as control knowledge for effective hand-tailored planning (Bacchus and Kabanza 2000; Doherty and Kvarnström 2001). The PDDL3 language (Gerevini et al. 2009) features temporal formulas (among others) in the role of *state constraints*. Work since then has devised compilation techniques (Edelkamp 2006; Baier and McIlraith 2006; De Giacomo, De Masellis, and Montali 2014; Torres and Baier 2015), and investigated how to effectively deal with (soft-goal) temporal plan preferences (Baier, Bacchus, and McIlraith 2007, 2009).

Here we focus on the simpler case where the state constraint is a non-temporal formula $\phi$ that must be false throughout the plan. We refer to such constraints as *avoid conditions*. This special case is relevant as avoid conditions naturally characterize dangerous or highly undesirable situations. For example, an avoid condition can capture states the user knows to be dead ends; or risky states in a deterministic approximation of a probabilistic planning application.

Avoid conditions can be trivially compiled into preconditions, but this incurs a large overhead and is, as our experiments illustrate, often not effective. Our contribution consists in advanced algorithmic methods. Apart from several compilation techniques, we adapt prior work in classical planning to design a method *learning* from the avoid condition during search, and a method using *abstraction* to predict states starting from which $\phi$ cannot be avoided.

Our learning method is based on so-called *trap learning*. Traps (Lipovetzky, Muise, and Geffner 2016) are sets of states from which there is no path to the goal. One can generalize from traps—and thus prune future search states—by minimizing the states to retain only the core reason why an escape is not possible. By initiating trap refinements from dead-end states encountered during forward search, one can incrementally extend the traps as a form of nogood learning (Steinmetz and Hoffmann 2017a). We change the original trap definition to take into account $\phi$, and show how trap learning can be adapted accordingly.

Our abstraction method is a form of state abstraction, a wide-spread method used to design heuristic functions in planning (Edelkamp 2001; Helmert et al. 2014; Seipp and Helmert 2018). Abstract state spaces group concrete states $s$ into block states $A$. Observe that, given such an abstract state space, we know that $s \in A$ can be pruned if all paths from $A$ to a goal block traverse a block $A'$ where $s' \models \phi$ for all $s' \in A'$. In other words, given an abstraction, we can *predict* that every plan for a state $s$ will necessarily traverse the avoid condition. The question remains how to tailor abstractions for this purpose. To this end, we leverage so-called Cartesian abstractions and their associated *counter-example guided abstraction refinement (CEGAR)* process (Seipp and Helmert 2013, 2018). We modify the CEGAR process to incorporate $\phi$ as an additional source of counter-examples and, therewith, of refinement steps.

We run a large-scale experiment on satisficing planning, optimal planning, and proving unsolvability, evaluating compilations, learning, and abstraction. We do so on (a) reformulated standard benchmarks that incorporate aspects (more) naturally formulated as avoid conditions; (b) benchmarks involving road maps (or similar), where we systematically impose avoid conditions of the form "do not use particular combinations of road segments"; and (c) a small collection of benchmarks where we generated avoid conditions automatically using trap learning (which is a bit artificial and merely serves as a showcase). The results show that our new methods can be superior, in particular for proving unsolvability.

## Preliminaries

We consider classical planning tasks in FDR notation (Bäckström and Nebel 1995; Helmert 2006). A *planning task* is a tuple $\Pi = \langle \mathcal{V}, \mathcal{A}, \mathcal{I}, \mathcal{G} \rangle$. $\mathcal{V}$ is a set of *variables*, each $v \in \mathcal{V}$ has a finite domain $\mathcal{D}_v$. A *fact* is a variable assignment $p = \langle v, d \rangle$ for $v \in \mathcal{V}$ and $d \in \mathcal{D}_v$. The *initial state* $\mathcal{I}$ is a complete assignment of $\mathcal{V}$. The goal $\mathcal{G}$ is a partial assignment of $\mathcal{V}$. For a partial variable assignment $P$, $\mathcal{V}(P) \subseteq \mathcal{V}$ denotes the set of variables $v$ for which $P(v)$ is defined. For $v \in (\mathcal{V} \setminus \mathcal{V}(P))$, we write $P(v) = \bot$. $\mathcal{A}$ is a set of *actions*. Each action $a \in \mathcal{A}$ has a *precondition* $pre_a$ and an *effect* $eff_a$, both partial variable assignments, and a non-negative cost $c_a \in \mathbb{R}_0^+$. The *states* $\mathcal{S}$ of $\Pi$ are all complete variable assignments. An action $a$ is *applicable* in a state $s$ if $s(v) = pre_a(v)$ for all $v \in \mathcal{V}(pre_a)$. The result is given by $s[\![a]\!]$ where $s[\![a]\!](v) = eff_a(v)$ for all $v \in \mathcal{V}(eff_a)$, and $s[\![a]\!](v) = s(v)$ for the other variables. These definitions are extended to sequences of actions in an obvious manner. A *plan for $s$* is a sequence of actions $\pi$ that is applicable in $s$ and $s[\![\pi]\!](v) = \mathcal{G}(v)$ for all $v \in \mathcal{V}(\mathcal{G})$. An *optimal plan* is a plan with minimal summed up action cost. $s$ is called a *dead end* if there is no plan for $s$. A *plan for $\Pi$* is a plan for $\mathcal{I}$. $\Pi$ is called *unsolvable* if $\mathcal{I}$ is a dead end.

Throughout the paper, we treat partial variable assignments, conjunctions of facts, and sets of facts synonymously. For ease of presentation, we assume that every conjunction of facts calling for different values for the same variable implicitly simplifies to false. We consistently use $\psi$ to denote conjunctions, and $\phi$ for arbitrary propositional formulae. By $\Psi$ and $\Phi$ we refer to formulae in disjunctive normal form (DNF) without negation, and treat them synonymously to sets of conjunctions. For any propositional formula $\phi$, we denote by $[\phi] \subseteq \mathcal{S}$ the set of all states in which it is satisfied.

An *avoid condition* $\phi$ is an arbitrary propositional formula over facts. A plan $a_1, \ldots, a_n$ for $\Pi$ is called *$\phi$-compliant* if $\mathcal{I} \notin [\phi]$, and it holds for all $1 \leq i \leq n$ that $\mathcal{I}[\![a_1, \ldots, a_i]\!] \notin [\phi]$. An *optimal $\phi$-compliant plan* is a $\phi$-compliant plan with minimal action cost. We say that a state $s$ is *$\phi$-unsolvable* if there is no $\phi$-compliant plan for $s$.

## Compilations

Compiling avoid conditions into the planning task is straightforward in principle, but the naïve method is very ineffective so it is worth thinking this through more carefully. Furthermore, compilations for temporal plan constraints are well known and we address a special case here. Hence we evaluate three compilation methods in our experiments. All these compilations operate at the PDDL input level.

**Conditions Compilation** The first, and most straightforward, compilation ensures $\phi$-compliance by conjoining $\neg \phi$ to the preconditions of all actions and the goal. We denote by $\Pi^{\neg \phi}$ the resulting FDR planning task. Trivially, the plans of $\Pi^{\neg \phi}$ are the $\phi$-compliant plans of $\Pi$.

**LTL Compilation** Our second method uses existing tools for compiling temporal formulas into planning

tasks (Edelkamp 2006; Baier and McIlraith 2006). This usually works in two steps: (1) building an automaton representation of the formula, and (2) encoding this automaton into the planning task via additional state variables and actions. For our simple LTL formula $G\neg \phi$ (*always* not $\phi$), the automaton representation will always consist of exactly two locations. The initial location is accepting and has a self-loop conditioned by $\neg \phi$. The other location is not accepting, and is reached from the initial location if $\phi$ is satisfied. We denote by $\Pi^{LTL}$ the compilation of this automaton into $\Pi$. $\Pi^{LTL}$ enforces an update of the automaton location in between applications of actions from $\Pi$. The automaton "blocks" as soon as it leaves its accepting state. Discarding the automaton-related actions, the plans of $\Pi^{LTL}$ are exactly the $\phi$-compliant plans of $\Pi$. Moreover, plan optimality is not affected provided the newly introduced actions have 0 cost.

**Axiom Compilation** Both $\Pi^{\neg \phi}$ and $\Pi^{LTL}$ suffer from the use of $\neg \phi$, which causes a blow-up in FD's translator if $\phi$ is a DNF (because $\neg \phi$ is a CNF which the translator naïvely transforms into a DNF). This motivates our last compilation, which employs *derived predicates*, aka *axioms* (Hoffmann and Edelkamp 2005), to avoid that problem.

Axioms are defined by rules of the form $p \leftarrow \psi_p$. The fact $p$ must not be affected by any action, i.e., its truth value must be completely determined by the axioms. In the simple (non-recursive) form of axioms that we need for our compilation, $p$ is true in a state iff the state satisfies one of its associated rule conditions $\psi_p$. To enforce $\neg \phi$ with axioms, we introduce a rule (avoid) $\leftarrow \phi$, and conjoin $\neg$(avoid) to the precondition of every action and to the goal. We denote by $\Pi^{\mathcal{X}}$ the resulting FDR task with axioms.

## Trap Learning

The basic algorithm we assume for our new advanced techniques is forward search on $\Pi$, while pruning all states that satisfy $\phi$. On top of this simple baseline, in what follows we introduce methods that can identify additional $\phi$-unsolvable states: trap learning in this section, abstraction in the next. Both methods preserve completeness (returning a $\phi$-compliant plan if one exists) and optimality (returning an optimal $\phi$-compliant plan).

### Background: Traps

*Traps* have been originally proposed for pruning dead ends during search (Lipovetzky, Muise, and Geffner 2016). Formally, a trap is a set of states $T \subset \mathcal{S}$ that (T1) does not contain any goal state, and (T2) is closed under transitions. The two conditions together imply that every state in $T$ is a dead end. To make use of traps in practice, they must be represented in some compact form. Lipovetzky, Muise, and Geffner considered DNF formulae of small, fixed-size, conjunctions over facts without negation. Verifying whether $[\Psi]$ satisfies (T1) and (T2) for such formula $\Psi$ boils down to syntactic checks on individual elements $\psi \in \Psi$. For (T1) one can compare $\psi$ with $\mathcal{G}$ directly. The test of (T2) is based on the progression operator $Progress(\psi, a)$ – the conjunction of all facts necessarily true after applying $a$ to any state

in $[\psi]$. (T2) is satisfied if $Progress(\psi, a)$ implies $\Psi$ for all $\psi$ and $a$. The implication can be tested efficiently by syntactic comparison to the members in $\Psi$.

---

**Algorithm 1** Computation of $\psi_s$ for the trap update $\Psi'$ as discussed in the text. 2 – 4 ensure that $\Psi'$ satisfies (T2); 5 – 7 take care of (T1).

1: $\psi_s \leftarrow \top$ for all $s \in \hat{S}$
2: **while** there are $\psi_s, a$ s.t. $Progress(\psi_s, a) \not\Rightarrow \Psi'$ **do**
3:    $\psi_s \leftarrow \psi_s \wedge \langle v', s(v') \rangle$ for some $v' \notin \mathcal{V}(\psi_s)$
4: **end while**
5: **for all** $\psi_s$ s.t. $\psi_s \not\Rightarrow \neg\mathcal{G}$ **do**
6:    $\psi_s \leftarrow \psi_s \wedge \langle v, s(v) \rangle$ for some $v \in \mathcal{V}(\mathcal{G})$ s.t. $s(v) \neq \mathcal{G}(v)$
7: **end for**

---

Steinmetz and Hoffmann (2017a) presented a method to build such $\Psi$ from experience made during search. Search is started with using the empty trap, $\Psi := \bot$, and is terminated as soon as a goal state is found. $\Psi$ is updated whenever search has visited a set of states $\hat{S}$ such that $(\hat{S} \cap [\Psi]) = \emptyset$, yet all transitions that leave $\hat{S}$ go into $[\Psi]$. In other words, $([\Psi] \cup \hat{S})$ is a trap that is not fully represented by $\Psi$ yet. For details how such $\hat{S}$ are identified exactly, we refer to (Steinmetz and Hoffmann 2017a,b). The refinement then aims at finding for every $s \in \hat{S}$ some $\psi_s \subseteq s$ such that $\Psi' := \Psi \vee \bigvee_{s \in \hat{S}} \psi_s$ still represents a trap. Once found, $\Psi$ is replaced by $\Psi'$ and search is resumed. The update ensures that $([\Psi] \cup \hat{S}) \subseteq [\Psi']$, i.e., the trap becomes increasingly larger. Every state newly represented by $\Psi'$ besides those in $\hat{S}$ may lead to additional pruning in the remainder of the search. To achieve this generalization, Steinmetz and Hoffmann attempt to keep the size of the individual $\psi_s$ as small as possible, via the greedy procedure sketched in Algorithm 1.

## Tailoring To Avoid Condition

For the purpose of identifying $\phi$-unsolvable states, the original trap conditions are overly restrictive. We relax the conditions as follows. A set of states $T \subseteq \mathcal{S}$ is a *$\phi$-trap* if (T1') every goal state in $T$ satisfies $\phi$, and (T2') every transition that leaves $T$ either originates in a state that satisfies $\phi$, or goes into one that does. It is straightforward to show that these weaker conditions are still sufficient to ensure the intended property:

**Theorem 1.** *If $T$ is a $\phi$-trap, then every state in $T$ is $\phi$-unsolvable.*

As before, operationalizing on this notion necessitates a compact representation on which the $\phi$-trap conditions can be checked directly. We stick to the DNF representation $\Psi$ from above, and extend the construction methods accordingly. As central to these methods, one can still determine on a per-element basis whether the overall set $[\Psi]$ satisfies (T1') and (T2'):

**Theorem 2.** *Let $\Psi$ be a DNF formula over facts without negation. $[\Psi]$ is a $\phi$-trap if it holds for all $\psi \in \Psi$ that*

*(t1') $(\psi \wedge \mathcal{G}) \Rightarrow \phi$, and (t2') it holds for all $a \in \mathcal{A}$ that $Progress(\psi \wedge \neg\phi, a) \Rightarrow (\Psi \vee \phi)$.*

(t1') and (t2') are straightforward extensions of the corresponding per $\Psi$-element tests for the original trap definition. Unfortunately, however, the appearance of $\phi$ in the adapted conditions makes it a priori impossible to verify them via pure syntactic comparisons. Consider for example (t1'). To verify this condition we need to decide whether $\psi \wedge \mathcal{G} \wedge \neg\phi$ is satisfiable. Without assumptions on $\phi$, this test is generally **NP**-complete due to the complexity of propositional satisfiability.

To avoid having to translate each individual test into an (expensive) SAT query, we exploit a simple trick. Let $\Phi$ be the transformation of $\phi$ into a positive DNF formula. $[\Phi]$ is a $\phi$-trap as it trivially satisfies (T1') and (T2'). While $\Phi$ in itself does not carry any new information, it however greatly simplifies the verification of (t1') and (t2'). It should be noted that the transformation may come at the cost of an exponential blow up of the formula size. But considering the worst-case complexity, this cost will have to be paid at some point. Observe that when replacing $\phi$ by $\Phi$, both conditions decompose into efficiently computable trap implication tests. This is apparent for (t1'), which becomes $(\psi \wedge \mathcal{G}) \Rightarrow \Phi$. Regarding (t2'), we note that the structure of $\Phi$ allows to move $\neg\Phi$ out of the progression, resulting in two separate conditions: (t2'a) $(\psi \wedge pre_a) \Rightarrow \Phi$ or (t2'b) $Progress(\psi, a) \Rightarrow (\Psi \vee \Phi)$. Details are provided in an appendix.

With these simplifications at hand it becomes straightforward to adapt the existing trap construction methods towards generating $\phi$-traps instead. In particular, the trap learning algorithm can be used almost as is. To take into account (t2'b), it suffices to initialize $\Psi$ to $\Phi$. This initialization is valid as observed above. (t2'a) and (t1') map directly into additional loop conditions in lines 2 and 5 of Algorithm 1. The correctness of the construction follows via the same arguments already provided by Steinmetz and Hoffmann (2017a):

**Theorem 3.** *Trap learning with the mentioned modifications terminates eventually, and $\Psi$ remains a $\phi$-trap at all time.*

# Abstraction for Avoid-Prediction

We recall Cartesian abstractions and show how to tailor them to the identification of $\phi$-unsolvable states.

## Background: Cartesian Abstractions

An *abstraction* for $\Pi$ is an equivalence relation $\sim$ between the states $\mathcal{S}$. The *abstract states* $\mathcal{S}^\sim$ of $\sim$ are given by its equivalence classes. For state $s$, we denote by $[s]_\sim$ the equivalence class that contains $s$, and omit $\sim$ if it is clear from the context. The *abstract state space* associated with $\sim$ is the transition system $\Theta^\sim = \langle \mathcal{S}^\sim, \mathcal{T}^\sim, s_{\mathcal{I}}^{\widetilde{\sim}}, \mathcal{S}_{\mathcal{G}}^{\widetilde{\sim}} \rangle$ with *abstract initial state* $s_{\mathcal{I}}^{\widetilde{\sim}} = [\mathcal{I}]$ and *abstract goal states* $\mathcal{S}_{\mathcal{G}}^{\widetilde{\sim}} = \{[s] \mid s \in \mathcal{S}, \mathcal{G} \subseteq s\}$. The *abstract transitions* are given by $\mathcal{T}^\sim = \{\langle [s], a, [s[\![a]\!]] \rangle \mid s \in \mathcal{S}, a \in \mathcal{A}$ applicable in $s\}$.

Let the variables of $\Pi$ be $\mathcal{V} = \{v_1, \ldots, v_N\}$. *Cartesian abstractions* (Seipp and Helmert 2018) are abstractions whose abstract states are of the form $A_1 \times A_2 \times \cdots \times A_N$, where $A_i \subseteq \mathcal{D}_{v_i}$ for all $i$.

This structure makes Cartesian abstractions particularly suitable for a *counter-example guided refinement* loop (CEGAR): The construction starts with the trivial abstraction that contains just a single abstract state. One then iteratively splits an abstract state into two until the abstraction provides enough information, or some size limit is reached. Each refinement step starts with the extraction of an abstract solution, i.e., an abstract path $[s_0], a_1, [s_1], \ldots, a_n, [s_n]$ from the abstract initial state $[s_0] = s_{\mathcal{I}}^{\sim}$ to some abstract goal state $[s_n] \in \mathcal{S}_{\mathcal{G}}^{\sim}$. If no such path exists, then $\Pi$ must be unsolvable, and the refinement terminates. Otherwise, the corresponding concrete path $\underline{s_0}, a_1, \underline{s_1}, a_2, \ldots$ is computed by applying the actions successively, starting from $\underline{s_0} = \mathcal{I}$. The computation is stopped when one of the following conditions is satisfied:

(C1) Concrete and abstract state do not match: $[\underline{s_i}] \neq [s_i]$.

(C2) Action $a_i$ is not applicable in $\underline{s_{i-1}}$.

(C3) $\underline{s_n}$ does not satisfy the goal.

If not stopped, we have found a plan for $\Pi$ and the refinement terminates. Otherwise, the violated condition is used to split an abstract state, guaranteeing that the same error cannot occur in future iterations ($\uplus$ denotes disjoint set union):

(C1) $[s_{i-1}]$ is split into $[t_1] \uplus [t_2]$ such that $\underline{s_{i-1}} \in [t_2]$ and $[t_2]$ no longer has an abstract transition to $[s_i]$ via $a_i$.

(C2) $[s_{i-1}]$ is split into $[t_1] \uplus [t_2]$ such that $\underline{s_{i-1}} \in [t_2]$ and $[t_2]$ has no abstract transition via $a_i$.

(C3) $[s_n]$ is split into $[t_1] \uplus [t_2]$ such that $\underline{s_n} \in [t_2]$ and $[t_2]$ is no longer an abstract goal state.

The selection of $[t_1]$ and $[t_2]$ is done via simple syntactic checks. During the entire construction, a full representation of the abstract state space is maintained. After each split, this representation can be updated efficiently by "rewiring" transitions to $[t_1]$ and $[t_2]$. For full details, we refer to the work by Seipp and Helmert (2018). Once the abstract state space has been updated, a new abstract solution is extracted, and the whole process starts anew.

## Tailoring to Avoid Conditions

An abstract state $[s]$ *implies* $\phi$, written $[s] \Rightarrow \phi$, if all represented concrete states $s' \in [s]$ satisfy $\phi$. An analysis of the abstract state space with respect to this property can yield information about $\phi$-unsolvable states that do not satisfy $\phi$ themselves:

**Theorem 4.** *Let $[t]$ be any abstract state. If every path from $[t]$ to any abstract goal state visits some $[s]$ s.t. $[s] \Rightarrow \phi$, then every state represented by $[t]$ is $\phi$-unsolvable.*

We next show how to apply this observation to Cartesian abstractions. In particular, we detail how to implement the implication test $[s] \Rightarrow \phi$ for Cartesian states $[s]$. Moreover, we adapt the CEGAR approach from above to specifically construct abstractions for the purpose of identifying $\phi$-unsolvable states.

**Algorithm 2** Recursive method to decide whether a Cartesian state $[s] = A_1 \times \cdots \times A_N$ contains a concrete state that violates the positive DNF $\Phi$. Initially, $i = 0$.

> **if** $\Phi = \emptyset$ **then**
>     **return** true
> **end if**
> **if** $i = N + 1$ **then**
>     **return** false
> **end if**
> **if** $\exists d_i \in N_i$: $\langle v_i, d_i \rangle \notin \psi$ for all $\psi \in \Phi$ **then**
>     **return** $Unsat(i + 1, \{\psi \in \Phi \mid v_i \notin \mathcal{V}(\psi)\})$
> **else**
>     **for all** $d_i \in A_i$ **do**
>         **if** $Unsat(i + 1, \{\psi \in \Phi \mid \psi(v_i) \in \{\bot, d_i\}\})$ **then**
>             **return** true
>         **end if**
>     **end for**
>     **return** false
> **end if**

**Implication Test** Unfortunately, deciding whether $[s] \Rightarrow \phi$ for Cartesian abstractions is **NP**-hard in general. This becomes apparent if all state variables are Boolean. For the full Cartesian product $[s]$, $[s] \Rightarrow \phi$ then holds exactly if $\phi$ is a tautology. Deciding the latter is known to be **NP**-complete. Yet despite the worst-case complexity, the implication check was usually not the bottleneck in our experiments. Our implementation runs a simple backtracking search for a state $t \in [s]$ such that $t \models \neg\phi$. Clearly, such state exists iff $[s] \Rightarrow \phi$ does not hold. Algorithm 2 shows the pseudo-code. It assumes a positive DNF representation $\Phi$ of $\phi$. Each recursive call obtains the index of the variable to assign next, as well as the conjunctions in $\Phi$ that can still be satisfied given what has been chosen so far. The method only tries multiple values if no single value can be chosen that rules out all remaining conjunctions in which the variable appears.

**CEGAR** To foster the creation of abstract states $[t]$ as in Theorem 4, we propose two separate extensions of the CEGAR approach, differing in how exactly $\phi$ triggers refinements. Both variants start with an abstract goal path $[s_0], a_1, [s_1], \ldots, a_n, [s_n]$ such that $[s_i] \not\Rightarrow \phi$ holds at all time. If such a path does not exist, then the abstraction already proves that $\mathcal{I}$ is $\phi$-unsolvable. We stop immediately.

The first variant straightforwardly extends the original abstract path analysis steps by the additional error condition:

(C4) The concrete state $\underline{s_i}$ satisfies $\phi$.

Whenever (C4) is satisfied, then due to above's restriction of the abstract goal paths, $[s_i]$ must represent states that satisfy $\phi$ as well as ones that do not. We split $[s_i]$ in a way that allows to distinguish those states within the abstraction. In particular, the refinement guarantees that $\underline{s_i}$ will map into $[s_i] \Rightarrow \phi$, which for one ensures progress in the construction (the same error cannot occur again in the future), and for another contributes towards satisfying Theorem 4 by removing an abstract path violating the prerequisites. More concretely, $[s_i]$ is split into abstract states $[t_1] \uplus \cdots \uplus [t_k]$ such

that $[t_k] \Rightarrow \phi$, and $\underline{s_i} \in [t_k]$. Contrary to the previous refinement steps, a split into exactly two abstract states ($k = 2$) is not possible in general. To illustrate this, let $x$ and $y$ be two binary variables, and $\phi = (x = 1 \land y = 1)$, and consider the abstract state $[s_i] = (\{0,1\} \times \{0,1\})$. $[t_k] \Rightarrow \phi$ can only be satisfied for $[t_k] = (\{1\} \times \{1\})$. However, every possible split of $[s_i]$ into this $[t_k]$ requires $k \geq 3$ abstract states.

We use the following procedure to find $[t_1] \uplus \cdots \uplus [t_k]$. We start with $[t] = [s_i]$ and $j = 1$. Let $v$ be any variable whose value set $A$ in $[t]$ is not a singleton. $[t]$ is split into two abstract states $[t_j]$ and $[t']$ by dividing $A$ into $(A \setminus \{\underline{s_i}(v)\})$ and $\{\underline{s_i}(v)\}$ respectively. If after the split $[t'] \Rightarrow \phi$ holds, then we are done. Otherwise $[t']$ becomes the new $[t]$, and we repeat. Since $[t']$ will become $\{\underline{s_i}\}$ eventually, the method is guaranteed to terminate with the desired result. Moreover, since abstract states are still iteratively split into pairs, the abstract state space can be updated in the same fashion as before.

To prioritize refinements based on $\phi$, our second variant attempts to check whether any abstract state $[s_i]$ on the path contains some concrete state $s \models \phi$, prior to conducting the original analysis steps. If such abstract state $[s_i]$ exists, then $[s_i]$ is split as above, and skip the original analysis steps altogether. Since this condition is checked before constructing the concrete path, the state $s \in [s_i]$ with $s \models \phi$ must be searched for actively. This is computationally more expensive than the simple check in (C4).

Figure 1: Illustration of an abstract state space. Self loops are omitted. The planning task consists of binary variables $x, y, z$, initially all 0, goal $z=1$, and three actions with $pre/eff$: ($a_1$) $y=0/y=1$; ($a_2$) $y=1/x=1$; and ($a_3$) $x=1/z=1$. The abstract states are depicted in terms of $A_x \times A_y \times A_z$. The avoid condition is $\phi = (y=1)$. Abstract states with dashed borders contain a state that satisfies $\phi$. Goal states have double borders.

We close the discussion with the remark that the abstraction can in general not be refined based on $\phi$ exclusively, i.e., (C1)–(C3) remain necessary. Consider the example in Figure 1. As the (spurious) path $[s_0], a_3, [s_1]$ shows, paths in the abstraction can simply bypass $\phi$ even if the concrete paths cannot. Note that this abstract path violates (C2). The corresponding refinement will split $[s_0]$ by dividing the values of $x$ into $\{0\}$ and $\{1\}$. This suffices to make all abstract goal paths pass through $[s_2] \Rightarrow \phi$, proving that no $\phi$-compliant plan exists.

## Experiments

We implemented all described methods in Fast Downward (FD) (Helmert 2006). The avoid condition is specified as an additional input file in the full PDDL *condition* syntax. The compilations are implemented as part of FD's translator component. All DNF conversions are done as part of the standard FD preprocessing. The experiments were run on machines with an Intel Xeon E5-2650v3 processor, and cutoffs of 30 minutes and 4 GB memory.

We conducted experiments in optimal and satisficing planning, as well as proving unsolvability. For each category, we chose a canonical base planner configuration: optimal planning via $A^*$ search with LM-cut (Helmert and Domshlak 2009); satisficing planning via greedy best-first search with two open lists and preferred operators using $h^{FF}$ (Hoffmann and Nebel 2001); and proving unsolvability via depth-first search with $h^{max}$ (Haslum and Geffner 2000) for dead-end detection. We extended these base configurations by the following prediction methods: "–" no prediction, only prune by $\phi$; "trap" $\phi$-trap learning; "A" Cartesian **a**bstraction constructed via the original CEGAR approach; "PA" **p**redicting **a**bstraction with the additional (C4) check; and "SPA" **p**redicting **a**bstraction with the more **s**trict $\phi$ refinement check. We experimented with abstraction size limits of $N \in \{10k, 20k, 40k, 80k, 160k\}$ abstract states. We also tested trap learning and Cartesian abstractions for pruning dead-ends in the $\Pi^{\neg\phi}$ and $\Pi^{LTL}$ compilations (not $\Pi^{\mathcal{X}}$ because neither of them supports axioms). We next describe our benchmarks, then discuss the results.

### Benchmark Design

Benchmarks with avoid conditions already appeared in IPC 2006 (Dimopoulos et al. 2006), encoded via state constraints. But hard state constraints were only used in benchmarks of the temporal track, and the constraints themselves heavily relied on temporal operators, which makes them unsuited for our experiments. Instead we created a new benchmark set, including solvable as well as unsolvable instances. We designed three categories of benchmarks.

**The "-$\Phi$" benchmarks.** Several well-known benchmarks actually already use avoid conditions, not modeled explicitly but instead encoded into complex precondition and/or effect-condition formulas. We have identified 6 such domains, and manually separated the avoid condition from the action descriptions in an equivalence-preserving manner. The domains and avoid conditions are: CaveDiving (IPC14), mutual exclusion relationship between some divers; Fridge, constraints on fridge components; Miconic, complex relationship between passengers allowed to be in the elevator simultaneously, legal elevator moves are restricted by boarded passengers; Nurikabe (IPC18), illegal groupings of board cells; Openstacks (IPC08), production and delivery must follow a particular order; Trucks (IPC06), relationship between the occupancy and location of truck storage areas. An explicit avoid condition is a natural model for all of these, and partly actually more natural than the original PDDL.

**The "-$\binom{n}{k}$" benchmarks.** These benchmarks add avoid conditions systematically to some standard benchmarks. The avoid conditions we added enforce an upper limit $k$ on the number of occurences of $n$ selected events in any plan. Specifically, we considered "road avoidance" in Storage, Transport, and Trucks, forcing each vehicle to not traverse

| | | Coverage | | | | | | | | | | | Search Reduction Factors (Left: Geometric Mean, Right: Maximum) | | | | | | | | | | | | | |
|---|---|---|---|---|---|---|---|---|---|---|---|---|---|---|---|---|---|---|---|---|---|---|---|---|---|---|
| | | Compilations | | | Pruning | | | | | | | | | | | | | | | | | | | | | |
| | | | | | | | A | | PA | | SPA | | | | A | | | | PA | | | | SPA | | | |
| Domain | # | $\Pi^{\neg\phi}$ | $\Pi^{\text{LTL}}$ | $\Pi^{\mathcal{X}}$ | – | Trap | 20k | 160k | 20k | 160k | 20k | 160k | Trap | | $20k$ | | $160k$ | | $20k$ | | $160k$ | | $20k$ | | $160k$ | |
| Satisficing | | | | | | | | | | | | | | | | | | | | | | | | | | |
| CaveDiving-Φ | 17 | **4** | **4** | **4** | **4** | 0 | **4** | **4** | **4** | **4** | **4** | **4** | | | 1.0 | 1.0 | 1.1 | 1.1 | 1.0 | 1.0 | 1.1 | 1.1 | 1.1 | 1.1 | 1.2 | 1.3 |
| Fridge-Φ | 24 | 1 | 6 | **22** | 21 | 21 | 21 | 21 | 21 | 21 | 21 | 21 | 1.0 | 1.0 | 1.0 | 1.0 | 1.0 | 1.0 | 1.0 | 1.0 | 1.0 | 1.0 | 1.0 | 1.0 | 1.0 | 1.0 |
| Miconic-Φ | 150 | 0 | 25 | 82 | 132 | **134** | 131 | 125 | 131 | 107 | 124 | 58 | 1.8 | 521.9 | 1.0 | 1.0 | 1.0 | 1.0 | 1.0 | 10.6 | 1.0 | 10.6 | 1.0 | 1.3 | 1.0 | 1.7 |
| Nurikabe-Φ | 20 | 0 | 2 | **13** | 12 | 11 | 11 | 9 | 11 | 8 | 11 | 6 | 1.0 | 1.1 | 1.0 | 1.0 | 1.0 | 1.0 | 1.0 | 1.0 | 1.0 | 1.0 | 1.0 | 1.1 | 1.0 | 1.1 |
| Openstacks-Φ | 30 | 0 | 1 | **30** | **30** | **30** | 21 | 18 | 19 | 18 | **30** | **30** | 1.0 | 1.0 | 1.0 | 1.0 | 1.0 | 1.0 | 1.0 | 1.0 | 1.0 | 1.0 | 1.0 | 1.0 | 1.0 | 1.0 |
| Trucks-Φ | 30 | 6 | 20 | 20 | **22** | 16 | **22** | 18 | **22** | 12 | **22** | 9 | 1.1 | 1.3 | 1.0 | 1.0 | 1.0 | 1.0 | 1.0 | 1.0 | 1.0 | 1.0 | 1.0 | 1.0 | 1.0 | 1.0 |
| $\sum$ Φ | 271 | 11 | 58 | 171 | **221** | 212 | 210 | 195 | 208 | 170 | 212 | 128 | 1.4 | 521.9 | 1.0 | 1.0 | 1.0 | 1.1 | 1.0 | 10.6 | 1.0 | 10.6 | 1.0 | 1.3 | 1.0 | 1.7 |
| Rovers-$\binom{n}{k}$ | 25 | 5 | 10 | 17 | **18** | 17 | **18** | 13 | **18** | 13 | 16 | 12 | 1.0 | 1.0 | 1.0 | 1.0 | 1.0 | 1.0 | 1.0 | 1.0 | 1.0 | 1.0 | 1.0 | 1.0 | 1.0 | 1.0 |
| Satellite-$\binom{n}{k}$ | 34 | 4 | 14 | 14 | **18** | 17 | 17 | 9 | 17 | 9 | 17 | 9 | 1.0 | 1.0 | 1.0 | 1.0 | 1.0 | 1.0 | 1.0 | 1.0 | 1.0 | 1.0 | 1.0 | 1.0 | 1.0 | 1.0 |
| Storage-$\binom{n}{k}$ | 28 | 0 | 9 | **12** | **12** | **12** | **12** | 10 | **12** | 10 | **12** | 10 | 1.0 | 1.0 | 1.0 | 1.0 | 1.0 | 1.0 | 1.0 | 1.0 | 1.0 | 1.0 | 1.0 | 1.0 | 1.0 | 1.0 |
| Transport-$\binom{n}{k}$ | 103 | 10 | 32 | 59 | **60** | **60** | **60** | 49 | **60** | 49 | **60** | 54 | 1.0 | 1.0 | 1.0 | 1.0 | 1.0 | 1.0 | 1.0 | 1.0 | 1.0 | 1.0 | 1.0 | 1.0 | 1.1 | 62.3 |
| Trucks-$\binom{n}{k}$ | 35 | 11 | 13 | 14 | 16 | 12 | 15 | 15 | 16 | 15 | **17** | **17** | 1.3 | 5.5 | 1.0 | 1.1 | 1.1 | 2.5 | 1.2 | 5.5 | 1.0 | 1.1 | 1.1 | 1.5 | 1.1 | 1.5 |
| $\sum \binom{n}{k}$ | 225 | 30 | 78 | 116 | **124** | 118 | 122 | 96 | 123 | 96 | 122 | 102 | 1.0 | 5.5 | 1.0 | 1.1 | 1.0 | 2.5 | 1.0 | 5.5 | 1.0 | 1.1 | 1.0 | 1.5 | 1.0 | 62.3 |
| Optimal | | | | | | | | | | | | | | | | | | | | | | | | | | |
| CaveDiving-Φ | 17 | **4** | **4** | | **4** | **4** | **4** | **4** | **4** | **4** | **4** | **4** | 1.0 | 1.0 | 1.0 | 1.0 | 1.0 | 1.0 | 1.0 | 1.0 | 1.0 | 1.0 | 1.0 | 1.0 | 1.2 | 1.2 |
| Fridge-Φ | 24 | 1 | 6 | | **10** | **10** | **10** | **10** | **10** | **10** | **10** | **10** | 1.0 | 1.0 | 1.0 | 1.0 | 1.0 | 1.0 | 1.0 | 1.0 | 1.0 | 1.0 | 1.0 | 1.0 | 1.0 | 1.0 |
| Miconic-Φ | 150 | 0 | 25 | | 85 | **86** | 83 | 79 | 74 | 71 | 75 | 51 | 1.1 | 2.1 | 1.0 | 1.0 | 1.0 | 1.0 | 1.0 | 2.2 | 1.0 | 2.2 | 1.0 | 1.3 | 1.0 | 1.3 |
| Nurikabe-Φ | 20 | 0 | 2 | | **10** | **10** | **10** | 9 | **10** | 8 | **10** | 6 | 1.0 | 1.1 | 1.0 | 1.0 | 1.0 | 1.0 | 1.0 | 1.1 | 1.0 | 1.1 | 1.0 | 1.3 | 1.0 | 1.3 |
| Openstacks-Φ | 30 | 0 | 1 | | **15** | **15** | **15** | **15** | **15** | 13 | **15** | **15** | 1.0 | 1.0 | 1.0 | 1.0 | 1.0 | 1.0 | 1.0 | 1.0 | 1.0 | 1.0 | 1.0 | 1.0 | 1.0 | 1.0 |
| Trucks-Φ | 30 | 3 | **10** | | **10** | **10** | **10** | **10** | **10** | **10** | **10** | 9 | 1.0 | 1.2 | 1.0 | 1.0 | 1.0 | 1.0 | 1.0 | 1.0 | 1.0 | 1.0 | 1.0 | 1.0 | 1.0 | 1.0 |
| $\sum$ Φ | 271 | 8 | 48 | | 134 | **135** | 132 | 127 | 123 | 116 | 124 | 95 | 1.1 | 2.1 | 1.0 | 1.0 | 1.0 | 1.0 | 1.0 | 2.2 | 1.0 | 2.2 | 1.0 | 1.3 | 1.0 | 1.3 |
| Rovers-$\binom{n}{k}$ | 25 | 2 | 3 | | **5** | 4 | **5** | **5** | **5** | **5** | **5** | **5** | 1.0 | 1.0 | 1.0 | 1.0 | 1.0 | 1.0 | 1.0 | 1.0 | 1.0 | 1.0 | 1.0 | 1.0 | 1.0 | 1.0 |
| Satellite-$\binom{n}{k}$ | 34 | 3 | 3 | | **4** | **4** | **4** | **4** | **4** | **4** | **4** | **4** | 1.0 | 1.0 | 1.0 | 1.0 | 1.0 | 1.0 | 1.0 | 1.0 | 1.0 | 1.0 | 1.0 | 1.0 | 1.0 | 1.0 |
| Storage-$\binom{n}{k}$ | 28 | 0 | **9** | | **9** | **9** | **9** | **9** | **9** | **9** | **9** | **9** | 1.0 | 1.0 | 1.0 | 1.0 | 1.0 | 1.0 | 1.0 | 1.0 | 1.0 | 1.0 | 1.0 | 1.0 | 1.0 | 1.0 |
| Transport-$\binom{n}{k}$ | 103 | 6 | 16 | | **17** | **17** | **17** | **17** | **17** | **17** | **17** | **17** | 1.0 | 1.0 | 1.0 | 1.0 | 1.0 | 1.0 | 1.0 | 1.0 | 1.0 | 1.0 | 1.0 | 1.0 | 1.0 | 1.0 |
| Trucks-$\binom{n}{k}$ | 35 | 7 | 8 | | 9 | 8 | 9 | 9 | 9 | 8 | **10** | **10** | 1.1 | 1.6 | 1.0 | 1.0 | 1.0 | 1.0 | 1.0 | 1.1 | 1.0 | 1.1 | 1.0 | 1.1 | 1.0 | 1.1 |
| $\sum \binom{n}{k}$ | 225 | 18 | 39 | | 44 | 42 | 44 | 44 | 44 | 43 | **45** | **45** | 1.0 | 1.6 | 1.0 | 1.0 | 1.0 | 1.0 | 1.0 | 1.1 | 1.0 | 1.1 | 1.0 | 1.1 | 1.0 | 1.1 |
| Unsolvability | | | | | | | | | | | | | | | | | | | | | | | | | | |
| Rovers-$\binom{n}{k}$ | 28 | 5 | 5 | 5 | 5 | 7 | 5 | 5 | 6 | **8** | 7 | **8** | 13.9 | 227.9 | 1.0 | 1.0 | 1.0 | 1.0 | 570.9 | 9.3K | 12.8K | 5.7M | 12.8K | 5.7M | 12.8K | 5.7M |
| Satellite-$\binom{n}{k}$ | 36 | 4 | 4 | 4 | 4 | 4 | 4 | 4 | **5** | **5** | **5** | **5** | 4.0 | 16.2 | 1.0 | 1.0 | 1.0 | 1.0 | 510.0 | 85.2M | 547.8 | 85.2M | 539.1 | 85.2M | 5.9K | 85.2M |
| Storage-$\binom{n}{k}$ | 28 | 0 | 8 | **16** | **16** | 15 | **16** | 10 | **16** | 14 | **16** | 11 | 1.0 | 1.1 | 1.0 | 1.0 | 1.0 | 1.0 | 36.8 | 7.2K | 266.7 | 45.7K | 5.0 | 102.0 | 42.1 | 7.2K |
| Transport-$\binom{n}{k}$ | 105 | 4 | 26 | 65 | 66 | 15 | 66 | 47 | 66 | 61 | **68** | 63 | 1.0 | 1.1 | 1.0 | 1.0 | 1.0 | 1.0 | 107.5 | 14.9M | 701.0 | 14.9M | 16.2 | 14.9M | 196.0 | 14.9M |
| Trucks-$\binom{n}{k}$ | 35 | 7 | 16 | 17 | 17 | 7 | 17 | 15 | 18 | **19** | **19** | **19** | 1.2 | 2.0 | 1.0 | 1.1 | 1.0 | 1.0 | 49.1K | 12.2M | 192.2K | 12.2M | 49.2K | 12.2M | 192.2K | 12.2M |
| $\sum \binom{n}{k}$ | 232 | 20 | 59 | 107 | 108 | 48 | 108 | 81 | 111 | 107 | **115** | 106 | 1.6 | 227.9 | 1.0 | 1.1 | 1.0 | 1.0 | 276.9 | 85.2M | 1.8K | 85.2M | 76.1 | 85.2M | 790.8 | 85.2M |
| R-NoMystery-T | 150 | 0 | 4 | 52 | 64 | **136** | 65 | 83 | 67 | 84 | 81 | 104 | 310.9 | 7.2K | 1.1K | 83.2M | 625.9K | 94.4M | 1.1K | 83.2M | 627.4K | 94.4M | 96.6K | 85.0M | 2.2M | 94.4M |
| R-Rovers-T | 150 | 0 | 4 | 7 | 8 | **122** | 9 | 7 | 9 | 7 | 9 | 8 | 578.8 | 20.9K | 1.0 | 1.0 | 101.7 | 43.4K | 1.0 | 1.0 | 485.8 | 227.4K | 1.0 | 1.1 | 5.4K | 18.6M |
| R-TPP-T | 25 | 0 | 0 | 8 | 16 | **20** | 17 | 15 | 17 | 14 | 18 | 17 | 171.9 | 978.8 | 2.7 | 5.9M | 3.1 | 5.9M | 2.7 | 5.9M | 3.4 | 5.9M | 79.3 | 12.0M | 111.1K | 79.2M |
| $\sum$ T | 325 | 0 | 8 | 67 | 88 | **278** | 91 | 105 | 93 | 105 | 108 | 129 | 299.1 | 20.9K | 197.1 | 83.2M | 40.8K | 94.4M | 198.1 | 83.2M | 52.9K | 94.4M | 9.4K | 85.0M | 811.3K | 94.4M |

Table 1: Coverage results are on the left, best in **bold**. Results for the compilations are shown for the base configurations only. The configuration names are described in the text. The right hand side shows the ratio of visited states without $\phi$-prediction ("–") by visited states with $\phi$-prediction. $K$ stands for thousand, $M$ for million. Larger values indicate more pruning. For each method per-domain geometric mean and maximum values are shown. Values across different configurations are not directly comparable due to a different instance basis.

$\geq k$ of $n$ selected connections in the road-map graph; and "same-achiever avoidance" in Rovers and Satellite, forcing each rover/satellite to not achieve $\geq k$ of $n$ selected goals.

We generated the benchmark instances as follows. The size of the avoid conditions scales with $\binom{n}{k}$, so to keep the size under control we fixed $k$ to 2 throughout. The $n$ road-map connections/goals are selected arbitrarily. For each base instance, we determined the smallest value of $n$, denoted $n_\infty$, for which no $\phi$-compliant plan exists. Where such an $n_\infty$ was found, we added the instance with avoid condition for $n_\infty - 1$ to the solvable benchmark set, and for $n_\infty$ to the unsolvable benchmark set. If $n_\infty = 2$, we only added the instance for $n = 2$ to the unsolvable set.

Note that these avoid conditions are DNF formulas. We acknowledge that this creates a bias in our benchmark set to DNF avoid conditions. It appears natural though for an avoid condition to take the form of a list of bad things that should not happen, which is a DNF if each "bad thing" is characterized conjunctively just like preconditions and the goal.

**The "-T" benchmarks.** Finally, we designed a small set of benchmarks using trap learning as an avoid-condition generator. We considered unsolvable resource-constrained benchmarks (Nakhost, Hoffmann, and Müller 2012), where trap learning empirically works best (Steinmetz and Hoffmann 2017a). For each benchmark instance, we use trap learning to compute a complete trap, i. e., a DNF $\Psi_\infty$ that proves the instance unsolvable. For generating the avoid condition, we then select the first $20\%$ of the conjunctions added to $\Psi_\infty$.

The advantage of this scheme is that it allows systematic benchmark generation; the disadvantage is that it is somewhat artificial, as the generated avoid conditions presumably are quite different from what a human user would specify. Our results on these benchmarks should thus be interpreted with care, and are included merely as a showcase.

## Results using Compilations

Consider Table 1. For the compilations, the results in the different categories (satisficing, optimal, and unsolvability) are qualitatively similar. Using additional dead-end detectors (trap learning/abstraction) on top of the compilations turned out to be detrimental in all cases, so we omit these results.

Both $\Pi^{\neg\phi}$ and $\Pi^{\text{LTL}}$ cause a significant overhead in grounding for almost all domains. This was to be expected for the $\binom{n}{k}$ and $\mathsf{T}$ part, as grounding in both compilations requires the conversion of the CNF $\neg\phi$ back into DNF, which with the standard FD translator method is exponential in the size of $\phi$. That said, the results are not much better on the $\Phi$ benchmarks either. This is because, after the elimination of existential quantifiers, the avoid conditions there turn into big disjunctions too (reinforcing our view that DNF appears to be a natural form of avoid condition). The results for $\Pi^{\neg\phi}$ are significantly worse than for $\Pi^{\text{LTL}}$ because the former needs to do the DNF conversion for every action, while the automaton construction in $\Pi^{\neg\phi}$ requires this only once.

The axiom compilation $\Pi^{\mathcal{X}}$ is designed to avoid these problems ($\Pi^{\mathcal{X}}$ is missing in the optimal part since axioms are not supported by the optimal planner configuration). Nevertheless, planning performance does not benefit from having the avoid condition encoded directly in the model. $\Pi^{\mathcal{X}}$ is dominated almost universally by the simple $\phi$-pruning baseline (pruning states that satisfy $\phi$).

## Results using Prediction Methods

For the $\phi$-prediction methods, Table 1 also shows search space size reduction statistics. We selected abstraction size limits of $20k$ and $160k$ whose results are representative.

The trade-off between overhead and benefit generally becomes better the more the solution space is constrained. In satisficing planning, coverage could be improved over the base configuration in two domains. While search effort could be reduced in other domains as well, this reduction does not outweigh the incurred overhead. Hence our prediction methods lag behind in terms of total coverage here. In optimal planning, coverage improvements are still limited to the same domains. However, as the overall search becomes more expensive, investing time into the predictor computation becomes (relatively) less of an issue. The impact of the prediction methods becomes clearest in proving unsolvability, where pruning is most important. Here coverage results are in favor of the prediction configurations in all but one domain.

For both satisficing and optimal planning, the impact of $\phi$-prediction in terms of search reduction highly varies between domains. In the $\Phi$ and $\binom{n}{k}$ benchmarks, there are many domains where the additional pruning has (almost) no effect. The reason lies in the structure of these domains. In

Openstacks-$\Phi$, avoiding $\phi$ is always possible if it is not satisfied already. In the Nurikabe-$\Phi$ instances, it is almost not possible to make an illegal group assignment. Similarly, in most of the $\binom{n}{k}$ domains, there usually exist enough alternatives to get around the avoid condition. In Rovers-$\binom{n}{k}$ and Satellite-$\binom{n}{k}$, all goals can usually be achieved by all the agents so that restrictions on which agent is used for which goal are not important. In the other $\binom{n}{k}$ domains, the road network is often highly connected, always leaving open an alternative route.

The positive examples are CaveDiving-$\Phi$, Miconic-$\Phi$, and Trucks in both variations. In these domains, wrong decisions early on can make $\phi$ unavoidable. For example, in CaveDiving-$\Phi$, different divers must assist each other at different stages. Starting with a wrong diver can make this is impossible. In Miconic-$\Phi$, boarding passengers in the wrong order can make it impossible to move the elevator later on without violating some of the constraints.

In the unsolvable benchmarks, reasoning over $\phi$ is most important, and the potential of $\phi$-prediction can be best seen. Comparing $\phi$-trap learning vs. the abstractions, each approach shows good results in some domains. In the $\binom{n}{k}$ domains, $\phi$-trap learning causes the larger overhead, while the abstractions provide better predictions. The superior results of $\phi$-trap learning in the $\mathsf{T}$ benchmarks must be treated with caution due to the benchmark design, as the avoid conditions are generated by a trap-learning process in the first place.

The results for unsolvable benchmarks also clearly show that our modifications of CEGAR have the intended effect. Both PA and SPA were often able to prove the initial state unsolvable directly (no search needed). The $\phi$ implication checks can slow down the abstraction construction though. This is visible in the cases where $\phi$-prediction was not that useful. Between the two CEGAR variants PA and SPA, there is no clear winner. Prioritizing refinements based on $\phi$ works better in some domains, focusing on abstract-transition flaws in others. Overall SPA provides slightly better pruning capabilities, but is also often more expensive in the construction.

## Conclusion

State constraints are a natural modeling construct in planning, and has so far been considered mostly in temporal form. Here we consider the non-temporal special case of avoid conditions $\phi$ that must be false throughout the plan. We have designed advanced methods predicting states unsolvable due to $\phi$, and our experiments show that they can pay off.

While our benchmarks are mostly designed having in mind a human modeller who specifies the avoid condition $\phi$, an interesting avenue for future research is to instead leverage this modeling construct to connect to offline domain analyses. Under-approximations of unsafe or dangerous regions of states naturally form avoid conditions. It may then make sense to consider non-deterministic or probabilistic planning, and to directly handle BDD representations of $\phi$.

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

## Proofs

**Theorem 2.** *Let $\Psi$ be a DNF formula over facts without negation. $[\Psi]$ is a $\phi$-trap if it holds for all $\psi \in \Psi$ that (t1') $(\psi \wedge \mathcal{G}) \Rightarrow \phi$, and (t2') it holds for all $a \in \mathcal{A}$ that $Progress(\psi \wedge \neg\phi, a) \Rightarrow (\Psi \vee \phi)$.*

*Proof.* Let $\psi \in \Psi$ be arbitrary. If $\psi$ satisfies (t1'), then clearly every goal state that satisfies $\psi$ must satisfy $\phi$ as well. In other words, the set of states $[\psi]$ satisfies (T1'). Given that $[\Psi]$ is simply the union over $[\psi]$ of all its elements, it therefore follows that $[\Psi]$ satisfies (T1').

The detailed definition of a general progression operator is not required for showing that (t2') implies (T2'). We refer the reader to e.g. (Rintanen 2008). Just recall that *Progress* guarantees for all states $s$, all actions $a$ applicable in $s$, and all $\phi_1$ that:

$$s \models \phi_1 \Rightarrow s[\![a]\!] \models Progress(\phi_1, a) \tag{1}$$

Let $s$ be any state that satisfies $\psi \wedge \neg\phi$. Let $a$ any action applicable in $s$. Then, $s[\![a]\!] \models Progress(\psi \wedge \neg\phi, a)$, and hence if $\psi$ satisfies (t2'), then $s[\![a]\!] \models (\Psi \vee \phi)$. In other words, every state that satisfies $\psi$ either satisfies $\phi$ as well, or every transition from this state must make true $\Psi \vee \phi$. It follows that the set of states $[\psi]$ satisfies (T2'). Again, $[\Psi]$ being the union of all these $[\psi]$, we conclude that $[\Psi]$ satisfies (T2'). □

**Theorem 5.** *Let $\Phi$ and $\Psi$ be DNF formulae over facts without negation. Let $a \in \mathcal{A}$ be any action. For every $\psi \in \Psi$, it holds that (t2') $Progress(\psi \wedge \neg\Phi, a) \Rightarrow (\Psi \vee \Phi)$ is satisfied iff one of (t2'a) $(\psi \wedge pre_a) \Rightarrow \Phi$ or (t2'b) $Progress(\psi, a) \Rightarrow (\Psi \vee \Phi)$ is satisfied.*

*Proof.* That (t2'a) and (t2'b) imply (t2') is obvious: if (t2'a) is satisfied, then $Progress(\psi \wedge \neg\Phi, a)$ becomes false, and hence (t2') holds trivially. Since $\psi \wedge \neg\Phi$ implies $\psi$, it also follows that $Progress(\psi \wedge \neg\Phi, a)$ implies $Progress(\psi, a)$. Hence via transitivity of implication, (t2') is satisfied if (t2'b) is satisfied.

To show the other direction, assume for contradiction that (t2') is satisfied but (t2'a) and (t2'b) are not. Let $\psi_0 = \psi \wedge pre_a$, and note that $Progress(\psi, a) = Progress(\psi_0, a)$. Consider the following truth assignment to facts:

$$\chi(\langle v, d \rangle) = \begin{cases} \top & \text{if } \langle v, d \rangle \in \psi_0 \\ \bot & \text{otherwise} \end{cases}$$

Since (t2'a) is violated, there can be no conjunction in $\Phi$ that is satisfied in $\chi$. Let $\chi'$ be the truth assignment to facts obtained by applying $a$ on $\psi$, i.e.,

$$\chi'(\langle v, d \rangle) = \begin{cases} \top & \text{if } \langle v, d \rangle \in \mathit{eff}_a \\ \bot & \text{if } \langle v, d \rangle \in \mathit{eff}_a \text{ and } d \neq d' \\ \chi(\langle v, d \rangle) & \text{otherwise} \end{cases}$$

Note that for every conjunction of facts $\psi'$, it holds that $\chi' \models \psi'$ if and only if $Progress(\psi, a)$ implies $\psi'$. Since $\chi \models \psi$ and $\chi \models \psi \wedge \neg\Phi$ as per our previous observation, it follows from the definition of progression (see Equation 1) that (1) $\chi' \models Progress(\psi, a)$ and (2) $\chi' \models Progress(\psi \wedge \neg\Phi, a)$. Due to (2) and the assumption that (t2') is satisfied, there must exist a conjunction $\psi'$ in $\Psi$ or $\Phi$ that is satisfied in $\chi'$. Due to (1), and according to our previous observation, $\psi'$ must then be implied by $Progress(\psi, a)$. This is in contradiction to the assumption that (t2'b) is violated. We conclude that one of (t2'a) or (t2'b) must be satisfied if (t2') is. □

**Theorem 3.** *Trap learning with the mentioned modifications terminates eventually, and $\Psi$ remains a $\phi$-trap at all time.*

*Proof.* The modified algorithm is depicted in Algorithm 3.

---

**Algorithm 3** Computation of $\psi_s$ for a $\Phi$-trap update $\Psi'$. $2 - 4$ ensure that $\Psi'$ satisfies (T2'); $5 - 7$ take care of (T1').

---

1: $\psi_s \leftarrow \top$ for all $s \in \hat{S}$
2: **while** there are $\psi_s$ and $a$ s.t. $(\psi_s \wedge pre_a) \not\Rightarrow \Phi$ and $Progress(\psi_s, a) \not\Rightarrow \Psi'$ **do**
3:     $\psi_s \leftarrow \psi_s \wedge \langle v', s(v') \rangle$ for some $v' \notin \mathcal{V}(\psi_s)$
4: **end while**
5: **for all** $\psi_s$ s.t. $(\psi_s \wedge \mathcal{G}) \not\Rightarrow \Phi$ **do**
6:     $\psi_s \leftarrow \psi_s \wedge \langle v, s(v) \rangle$ for some $v \in \mathcal{V}(\mathcal{G})$ s.t. $s(v) \neq \mathcal{G}(v)$
7: **end for**

---

We first show that whenever a refinement of $\Psi$ is started, it holds for the corresponding states $\hat{S}$ that $([\Psi] \cup \hat{S})$ is a $\phi$-trap. The refinement condition was left unchanged, i.e., a refinement is initiated whenever search visited states $\hat{S}$ such that (R1) $(\hat{S} \cap [\Psi]) = \emptyset$, and (R2) all transitions leaving $\hat{S}$ go into $[\Psi]$. We show below that $\Psi$ always remains a $\phi$-trap, provided that every refinement is seeded with a $\phi$-trap as input. Given that $[\Psi]$ is a $\phi$-trap, (R2) immediately implies that $(\hat{S} \cup [\Psi])$ satisfies (T2'). To show that (T1') is also satisfied, note that $\hat{S}$ must be disjoint from $[\mathcal{G}]$ as well as from $[\Phi]$. The latter is guaranteed by (R1) and the initialization of $\Psi$. $\hat{S}$ must be goal-disjoint because every goal state encountered in search until the point of the refinement must satisfy $\Phi$ (which are not in $\hat{S}$). Since $\hat{S}$ does not contain a goal state, and $[\Psi]$ must satisfy (T1'), it follows that their union still satisfies (T1'). In summary, $([\Psi] \cup \hat{S})$ is a $\phi$-trap.

We finally show that Algorithm 3 terminates and correctly returns a $\phi$-trap $\Psi'$. As we have shown above, $\hat{S}$ never contains a goal state. Therefore a variable as required in line 6 exists for every state in $\hat{S}$. Moreover, adding the corresponding fact to $\psi_s$ makes the conjunction of $\psi_s$ and $\mathcal{G}$ to become false, at which point $(\psi_s \wedge \mathcal{G}) \Rightarrow \Phi$ holds trivially. Therefore the overall loop must terminate eventually. Once terminated, (t1') obviously holds for all $\psi_s$.

If no variable requested in line 3 exists, then $\psi_s = s$. However, such $\psi_s$ can never satisfy the loop condition. For

$\psi_s = s$, the progression operation just enumerates the transitions leaving $s$. As per (R2), every such transition must go into $[\Psi]$ or back into $\hat{S}$, both are covered by $[\Psi']$. The same argument also explains why the overall while loop must terminate eventually. Once terminated, Theorem 5 guarantees that (t2′) holds for all $\psi_s$.

The bottom line is: if $[\Psi]$ was a $\phi$-trap before the refinement, $\hat{S}$ is such that (R1) and (R2) are satisfied, then $[\Psi']$ is a $\phi$-trap after the refinement. Since $\Psi$ is initialized to $\Phi$, $\Psi$ represents a $\phi$-trap initially. That $\hat{S}$ satisfies (R1) and (R2) is guaranteed by the refinement caller. We conclude that $\Psi$ remains a $\phi$-trap throughout the entire search period.

$\square$