# OpenReview forum: "Classical Planning with Avoid Conditions"
_icaps-conference.org/ICAPS/2021/Workshop/HSDIP — HSDIP 2021_

### Official Review · AnonReviewer2 · 2021-05-25

**Confidence:** 4
**Overall Score:** Accept

**Review:**

The paper introduces several ways to use so called avoid conditions in a search for a plan. An avoid condition is a DNF formula that should never be true along the execution of the plan. The author compare existing techniques (compilation to LTL, encoding in the preconditions, and encoding in axioms) and add two main new ideas: avoid conditions can be used inside the algorithms for discovering traps and in the CEGAR approach to compute Cartesian abstractions. in both cases the authors show how to consider avoid conditions and prove that the resulting search terminates with an optimal plan among those that avoid the given condition.

In an experimental section, the authors run the discussed algorithms on several domains in different settings. The impact of the new techniques is unfortunately often small compared to simple pruning using the avoid conditions. The main exception are domains in an unsolvability setting that were specifically designed to show the strengths of the algorithm.

The paper emphasizes that avoid conditions in DNF are the natural form to specify avoid conditions but does not give a good example for this. The only specific example I found was in transportation domains where the agent is allowed to use only k out of n roads. I do not see this as a particular "real world" scenario. If a road is risky to use, why would it be OK to use k of them? Also, why is it safer to use one risky road two times rather than two risky roads once? In my opinion, a more natural model would add a variable "number_of_risks_taken" to the problem and increase it with every risky operation, with a limit on that variable in every precondition. I think this is particularly relevant because in the paper k is fixed to 2, which means that only two risks can ever be taken. In the other domains, I understand even less what the avoid conditions are used for. More examples and motivation would certainly help here.

The paper is written clearly and the theoretical results are sound. Since the empirical results are not overwhelming, I see the theoretical results as the main contribution. The constructed domains show that there are at least some cases where the new techniques bring a significant benefit. This is clearly sufficient for acceptance at HSDIP.

---

### Official Review · AnonReviewer1 · 2021-05-26

**Confidence:** 4
**Overall Score:** Accept

**Review:**

The paper investigates planning tasks with avoid conditions, i.e. tasks where
we are given a formula $\varphi$ which must be avoided in all states. While
this is a special type of PDDL3 constraint (and thus existing PDDL3
compilations can handle it), an approach tailored for avoid conditions seems
more promising in terms of efficiency. The paper proposes two techniques to
predict states from which it is unavoidable to reach a state satisfying
$\varphi$, one based on Traps and one on CEGAR. The approach is evaluated on a
collection of satisfying, optimal and unsolvable benchmarks against existing
compilation and simple pruning of states that satisfy $\varphi$. The results show
that the new approaches are significantly better than PDDL3 compilation methods
and perform similar to pruning.

The topic of the paper is a good fit for HSDIP and the proposed prediction
methods are novel and very intriguing. While the results aren't fully
convincing yet (in that the prediction methods do not clearly outperform simple
pruning) they do show that the approach has potential and are thus convincing
enough for a workshop submission.

I am however not fully convinced about the correctness of the trap approach due
to two reasons:
1) I don't understand condition (2'): You say every transition that leaves the
   trap either originates in a state satisfying $\varphi$, or goes into one that
   does. So for each transition $\langle s,t,s' \rangle$ with $s \in T$ and $s' \not\in T$ we have
   either $s \models \varphi$ or $s' \models \varphi$ correct? But if $s' \models
   \varphi$ doesn't it follow that $s' \in T$?
2) I don't understand how exactly the refinement works and why it is correct.
   Which trap conditions does the refinement need to satisfy, (1) and (2) or
   (1') and (2')? From the text I assume its (1) and (2), but why did you
   introduce (1') and (2') then? But if it's (1') and (2') then I don't see how
   that guarantees correctness since (1') and (2') are weaker. Finally I don't
   understand how you concretely check whether a refinement satisfies condition
   (2) or (2').

(Due to these doubts I currently only recommend a weak accept, but if the
authors clarify the raised issues I'm more than happy to accept the paper
since I think the idea is very interesting and promising.)

AFTER AUTHOR RESPONSE:
I see now how this is correct and change my score to accept.


Minor comments:
 - Preliminaries: "A fact is variable ..." -> is a variable ...
 - Preliminaries: "The goal G is partial ... " -> is a partial
 - Preliminaries: "A plan for s a sequence --- " -> is a sequence
 - Tailoring to Avoid condition: "Say that an abstract state [s] implies $\phi$,
 ..., if all ..." -> We say that
 - References: Baier and McIlraith 2006 are missing page numbers
 - References: Hoffman and Edelkamp 2005: ipc -> IPC

---

### Decision · Program_Chairs · 2021-06-10

**Decision:**

Accept

**Comment:**

Congratulations, both reviewers clearly recommend to accept the paper. For the final version, we encourage you to elaborate on the Trap approach (AnonReviewer1) and the Miconic example (AnonReviewer2) if space permits.